# Chromosomal variations of *Lycoris* species revealed by FISH with rDNAs and centromeric histone H3 variant associated DNAs

**Mao-Sen Liu, Shih-Hsuan Tseng, Ching-Chi Tsai¤, Ting-Chu Chen, Mei-Chu Chung** *

Institute of Plant and Microbial Biology, Academia Sinica, Taipei, Taiwan

¤ Current address: CH Biotech R & D Co., LTD., Nantou, Taiwan
* bomchung@gate.sinica.edu.tw

## Abstract

*Lycoris* species have various chromosome numbers and karyotypes, but all have a constant total number of chromosome major arms. In addition to three fundamental types, including metacentric (M-), telocentric (T-), and acrocentric (A-) chromosomes, chromosomes in various morphology and size were also observed in natural populations. Both fusion and fission translocation have been considered as main mechanisms leading to the diverse karyotypes among *Lycoris* species, which suggests the centromere organization playing a role in such arrangements. We detected several chromosomal structure changes in *Lycoris* including centric fusion, inversion, gene amplification, and segment deletion by using fluorescence *in situ* hybridization (FISH) probing with rDNAs. An antibody against centromere specific histone H3 (CENH3) of *L. aurea* (2n = 14, 8M+6T) was raised and used to obtain CENH3-associated DNA sequences of *L. aurea* by chromatin immunoprecipitation (ChIP) cloning method. Immunostaining with anti-CENH3 antibody could label the centromeres of M-, T-, and A-type chromosomes. Immunostaining also revealed two centromeres on one T-type chromosome and a centromere on individual mini-chromosome. Among 10,000 ChIP clones, 500 clones which showed abundant in *L. aurea* genome by dot-blotting analysis were FISH mapped on chromosomes to examine their cytological distribution. Five of these 500 clones could generate intense FISH signals at centromeric region on M-type but not T-type chromosomes. FISH signals of these five clones rarely appeared on A-type chromosomes. The five ChIP clones showed similarity in DNA sequences and could generate similar but not identical distribution patterns of FISH signals on individual chromosomes. Furthermore, the distinct distribution patterns of FISH signals on each chromosome generated by these five ChIP clones allow to identify individual chromosome, which is considered difficult by conventional staining approaches. Our results suggest a different organization of centromeres of the three chromosome types in *Lycoris* species.

**Data Availability Statement:** All relevant data are within the paper and its Supporting Information files.

**Funding:** This work was funded by the Institute of Plant and Microbial Biology, Academia Sinica, Taiwan, ROC, and Ministry of Science and Technology, Taiwan, ROC (grant no. MOST 106-2313-B-001-007 -MY3) to MCC. The funders had no role in study design, data collection and analysis, decision to publish, or preparation of the manuscript.

**Competing interests:** The author, Ching-Chi Tsai is currently employed by CH Biotech R & D Co., LTD (CHB); but, before she moved to CHB, she worked as a post-doctor and joined this study which was supported by the funding from the Institute of Plant and Microbial Biology, Academia Sinica, Taiwan, ROC. That commercial company, CH Biotech R & D Co., LTD, did not play any role in this study. Therefore, the author's affiliation is "Institute of Plant and Microbial Biology, Academia Sinica, Taipei, Taiwan", and "CH Biotech R & D Co., LTD "is her current address. The authors have declared that no competing interests exist.

## Introduction

The genus *Lycoris* (Amaryllidaceae) which consists of about 20 species are important ornamental crops with medicinal value [1]. *Lycoris* species feature diverse chromosome number and morphology [2]. The chromosome numbers of *Lycoris* species range from 2n = 12 to 44, including diploid, triploid, tetraploid and aneuploid [3, 4]. The chromosome complement of *Lycoris* species may contain three fundamental types, including metacentric (M), telocentric (T), and acrocentric (A) chromosomes [2]. *Lycoris* taxa are grouped into A, MT, and MT-A karyotypes based on chromosome complements. *Lycoris* species of A or MT karyotypes are usually fertile diploids, whereas the MT-A karyotype is mainly sterile hybrids of MT- and A-karyotypes [5]. The chromosome numbers naturally show wide variation, but the total number of chromosome major arms (nombre fondamantal or NF) in all *Lycoris* species is always a multiple of 11 [6]. Both fusion and fission translocation have been considered as main mechanisms leading to the diverse karyotypes among *Lycoris* species, which suggests that centromere organization may play role in such arrangements.

The centromere is a specialized chromosomal structure for kinetochore formation and spindle microtubule attachment during meiotic and mitotic cell division, which is essential for faithful chromosome segregation and genome stability. Although the centromere function is highly conserved, the centromeric DNA sequences show little or no conservation among organisms [7]. Thus, centromeres are defined epigenetically by the presence of a centromere-specific histone H3 variant, called CENH3 [8–10]. CENH3 is incorporated into the nucleosome to replace canonical H3 in the centromeric region of most eukaryotic chromosomes [11–13]. Each of the reported CENH3 is a highly conserved protein with a common histone H3 core sequence and highly diverse N-terminal and loop-1 domains [14, 15].

The centromeres in most draft genome sequences usually present as incomplete or blank regions because the highly repetitive and very long span nature of centromeric DNA is still a challenge with current DNA sequencing technologies. CENH3 is a universal marker to identify active centromeres, so the DNA sequences associated with CENH3 have been conventionally isolated and characterized by using chromatin immunoprecipitation (ChIP) with an anti-CENH3 antibody, followed by ChIP-cloning and ChIP-sequencing (ChIP-seq) in many plants. The structure and function of CENH3 has been intensively studied in *Zea mays* [16], *Oryza* [17, 18], *Hordeum* [19], *Allium* [20], *Brassica* [21] and several members of the Leguminosae family [22–25]. In most cases, the centromeric sequences were identified as species-specific microsatellites, minisatellites, macrosatellites, and retrotransposons. These sequences were confirmed to locate at centromeric regions in these species by fluorescence *in situ* hybridization (FISH).

*Lycoris* species are known to have a giant genome size; for example, the 1Cx-value of *L. aurea* was estimated at 24.5 pg [26] or 30.40 pg [27]. The *Lycoris* genome assembly has not been released, and whole genome sequencing data are scarce. Only a transcriptome with an expressed sequence tag (EST) dataset for *L. aurea* [28] and two complete chloroplast genome sequences of *L. squamigera* [29] and *L. radiata* [30] have been reported. Among the plants with centromeric sequences reported, *Allium* species [20], also belonging to the Amaryllidaceae family, are the most phylogenetically close to *Lycoris* species.

To understand the role of the centromere in the karyotype diversification of *Lycoris* species, we aimed to identify *Lycoris* CENH3 variants and the centromeric DNA sequence. In this study, we isolated a putative *Lycoris CENH3* clone (*LaCENH3*) based on coding region sequence alignment of *Allium* species [20], then produced a peptide antibody against the deduced LaCENH3. The CENH3-associated DNA sequences of *L. aurea* were isolated by using a ChIP-cloning experiment and their cytological positions were confirmed by using

FISH. Our results suggest a different organization of centromeres among the three chromosome types in *Lycoris* species. We also report some chromosomal structure variations detected by FISH, which implies a new centromere formation following rearrangements. We discuss the role of centromere-related sequences in chromosomal structure variation.

## Materials and methods

### Plant materials

*Lycoris aurea* (2n = 14, 8M+6T), *L. radiata* (2n = 22, 22A) and an artificial hybrid *L. aurea × L. radiata* (2n = 18, 4M+3T+11A) were used in this study. The karyotype of a variant of triploid (2n = 33, 31A+1M'+1m) was also investigated. Plants were maintained in a greenhouse at Academia Sinica, Taipei. Bulbs were harvested for genomic DNA extraction. Young root tips were collected for RNA extraction, ChIP, and cytology experiments.

### Nucleotide extraction and reverse transcription

Bulbs and young root tips of *L. aurea* were harvested and grounded into powder in liquid nitrogen by using a mortar and a pestle for genomic DNA and total RNA extraction. Genomic DNA was extracted by use of the DNeasy Plant Mini kit (Qiagen, Hilden Germany). Total RNA was extracted by use of the RNease Plant Mini Kit (Qiagen, Hilden Germany) and possible DNA was removed by use of the TURBO DNA-*free* Kit (Invitrogen, Thermo Fisher Scientific, Waltham, MA, USA). The first-strand cDNA was synthesized by use of the ImProm-II Reverse Transcription System (Promega, Wisconsin USA).

### Putative LaCENH3 cloning and antibody generation

For cloning the coding region of *L. aurea* CENH3 (LaCENH3), a degenerated primer pair, LaCENH3-dF (5′-ATGGCGAGAACTAARCABAT-3′) and LaCENH3-dR (5′-TYAYGA AAAATGTCKWGMRCCA-3′), was designed based on *CENH3* coding region sequence alignment of *Allium fistulosum* (*AfiCENH3*: GenBank accession number AB571555.1), *A. cepa* (*AceCENH3*: GenBank accession number AB600275.1), *A. sativum* (*AsaCENH3*: GenBank accession number AB571556.1) and *A. tuberosum* (*AtuCENH3*: GenBank accession number AB571557.1) [20]. One clone showing high DNA sequence similarity and protein identity to *Allium* species was selected and assigned as *LaCENH3* (GenBank accession number MW036651).

The peptides (ARTKQTAQNHPSKRRRDAAS) were synthesized based on the deduced amino acid sequence (LaCENH3) of the putative coding region of *LaCENH3* from amino acid A1 to S20 at the N-terminus, which was used to produce a peptide antibody against LaCENH3 in rabbit serum. This anti-LaCENH3 antibody was used for further immunoassay and ChIP experiments.

### ChIP and ChIP-cloning

ChIP was performed following the instructions of the Universal Plant ChIP-seq Kit (Diagenode Diagnostics, Belgium) with some modifications. The young root tips of *L. aurea* were harvested and treated with crosslinking buffer containing 1% paraformaldehyde for 15 min under vacuum. Chromatin was extracted with extraction buffer supplemented with a protease inhibitor cocktail (Sigma-Aldrich, MO USA), β-mercaptoethanol, and phenylmethylsulfonyl-fluoride (PMSF, Sigma-Aldrich, MO USA). The extracted chromatin was sheared to fragments enriched around 100–700 bp by use of a Bioruptor Sonicator (Diagenode Diagnostics,

Belgium). LaCENH3-associated DNA fragments were immunoprecipitated by using the anti-LaCENH3 antibody.

ChIPed DNA fragments were released from LaCEN3 by treating with 0.01% (w/w) protenase K (10 mM EDTA, 1% SDS and 50 mM Tris-HCl, pH8) at 65°C for 4 h. The released ChIP DNA was end-repaired with T4 DNA polymerase (BioLabs, New England) and 3'-dA was added by using OneTaq DNA polymerase (BioLabs, New England) before being cloned into a pCRII vector (Invitrogen, USA). After transforming, the vectors were transferred into an *E. coli* strain ECOS X (Yeastern, Taiwan) and positive clones were selected after overnight culture on agar plates containing ampicillin. The size of insertions was confirmed by PCR with the M13 forward (5'-GTTGTAAAACGACGGCCAG-3') and M13 reverse (5'-ACACAGGAAA-CAGCTATGAC-3') primer pair.

## Dot blot analysis

ChIP clones with insert DNA fragments confirmed by colony PCR were further checked for genomic abundance by dot blot analysis. The PCR product (2 μL) of each clone was dotted onto a positively charged nylon transfer membrane (Amersham Hybond-N+, Global Life Sciences Solutions USA, Marlborough, MA, USA) and cross-linked by UV under 0.120 J/cm$^2$ in a Bio-Link Crosslinker BLX-254 (Witec AG, Sursee, Switzerland). Simultaneously, the plasmid DNA of *Lycoris* 5S rDNA (*La5S rDNA*: GenBank accession number MW036652) was used as a control. Genomic DNA of *L. aurea* was boiled for 30 min to obtain DNA fragments < 10 kb before being labeled with Digoxigenin-11-dUTP (Dig-dUTP; Roche, Germany) by using Nick Translation Mix (Roche, Germany). The membrane was incubated with Dig-labeled genomic DNA probes in hybridization buffer containing 5X Saline Sodium Citrate buffer (SSC), 5X Denhardt solution and 0.5% SDS. After hybridization overnight at 37°C, the membrane was washed with TBST buffer (20 mM Tris-HCl pH 7.6, 200 mM NaCl, and 0.1% Tween 20) and immune-detected with anti-Dioxigenin antibody (in 1:2500 dilution, Abcam, UK) as the primary antibody and anti-mouse IgG conjugate HRP antibody (1:5000 dilution, Abcam, UK) as the secondary antibody.

## Cytological immunostaining and FISH

Cytological immunostaining was conducted as previously described [20] with minor modifications. Young root tips were fixed for 30 min in freshly made 4% paraformaldehyde in phosphate-buffered saline (PBS: 137 mM NaCl, 27 mM KCl, 10 mM Na$_2$HPO$_4$, 18 mM KH$_2$PO$_4$, pH 7.4) with 0.2% Triton X-100, then washed twice with PBS before being stored at 4°C. The fixed root tips were macerated for 2 h at 37°C with an enzyme mixture of 2.0% pectinase (Sigma Chemical Co., St. Louis, MO, USA) and 2.0% cellulase Onozuka RS (Yakult Honsha Co., Tokyo) in PBS. Softened tissues were washed twice with PBS and squashed onto slides coated with poly-L-lysine (Sigma Chemical Co.). The slides were immersed in liquid nitrogen to remove the cover slips and dried on a 40°C hot plate for 30 min before use.

A blocking solution of 0.5% blocking reagent (Roche, Germany) in TNT buffer (100 mM Tris–HCl, pH 7.5; 150 mM NaCl, and 0.5% Tween 20) was applied to the slide and incubated in a humidified box at 37°C for 30 min. The rabbit anti-LaCENH3 antibody was diluted 1:100 in blocking solution, applied to the slides, and incubated at 37°C for 1 h. After two washes with TNT buffer for 2 min, slides were incubated with goat anti-rabbit fluorescein isothiocyanate–conjugated antibody (1:100 dilution in blocking solution) for 1 h. After two washes with TNT buffer for 2 min, the chromosomes were counterstained with 4', 6-diamidino-2-phenylindole (DAPI) in Vectashield (1.5 mg mL$^{-1}$; Vector Laboratories, USA).

Chromosome preparation and FISH experiments were performed following published protocols [31]. Chromosomes of an artificial hybrid *L. aurea* × *L. radiata* (2n = 18,4M+3T+11A) were used for FISH analysis, which would be convenient to check the specificity of each probe to M-, T-, and A-type chromosomes per FISH experiment. Briefly, chromosomes were prepared from the root tips by enzyme maceration and the flame dry method [31]. Plasmid DNA of selected ChIP-clones, 45S rDNA cloned from wheat [32] and 5S rDNA cloned from *L. aurea* (*La5SrDNA*), and telomeric sequence amplified with the sequence $(TTAGGG)_5$ as a primer in the absence of template [33] were labeled by nick translation by using a DIG- or Biotin-Nick Translation Mix (Roche, Germany). The Dig- and biotin-labeled probes were visualized by using rhodamine-conjugated anti-digoxigenin antibody (Roche, Germany) and FITC-conjugated anti-Biotin (Molecular Probes, USA), respectively. Chromosomes were counterstained with DAPI as described above for immunostaining. Fluorescent signals were captured and edited as described previously [33, 34]. For each examined probe, more than 20 metaphase cells with bright and reproducible signals were selected for analysis.

## Results

### FISH analysis of chromosomal structure variations of *Lycoris*

FISH results showed evidence of centromeric fusion in a triploid *Lycoris* plant. Among the triploid *Lycoris* population (2n = 33A) collected in Kinmen county, Taiwan, one plant had a large M-type chromosome (M') and a small M-type chromosome (m) in chromosome complement; its karyotype could be formulated as 2n = 33, 31A+1M'+m (Fig 1A). The distribution pattern of FISH signals of 45S rDNA and 5S rDNA (Fig 1B) on 31 A-type chromosomes, excluding M' and m, was identical to those 11 A-type chromosomes of *L. albiflora* (2n = 17, 5M + 1T + 11A) being reported previously [34]. *Lycoris albiflora* was considered as a hybrid between *L. traubii* (2n = 12, 10M + 2T) and *L. radiata var. radiata* (2n = 3x = 33, 33A) [35]. FISH results suggested that the triploid *Lycoris* plant investigated here was a chromosomal variant of *L. radiata var. radiata* which was an essential autotriploid (2n = 3x = 33, 33A). The m chromosome with 45S rDNA and telomeric sequence (Fig 1C) could be evenly segregated to daughter cells in anaphase (Fig 1D), so this m chromosome should be equipped with a functional centromere. FISH signals of 45S rDNA presented at the centromeric region of the M' and m chromosomes (Fig 1A–1C), which suggested that the M' and m chromosomes were products of fusion of two A-type chromosomes, one broken within the array of 45S rDNA repeats on chromosome 1 and another within the centromeric region of chromosome 8 (Fig 2). The M' chromosome contained two large chromosomal arms. The m chromosome contained a partial short arm with 45S rDNA locus of one A-type chromosome and a short arm with a centromere from another A-type chromosome (Fig 2).

One inversion M-type chromosome was found in an artificial hybrid *L. aurea* × *L. radiata* (2n = 18, 4M+3T+11A) based on the chromosomal distribution of 5S rDNA sites (Fig 3). As previously reported [34], one M-type chromosomes of *L. aurea* (n = 4M+3T) has three 5S rDNA loci, including one with weak FISH signal near the centromere (C-site) and two conspicuous signals, one at the intercalary region of the p-arm (P-site) and another at the distal end of the q-arm (Q-site; Fig 3A). In some cells, two conspicuous signals were detected on the q-arm on this M-type chromosome, which suggested a pericentric segment including the 5S rDNA locus at the p-arm (P-site) inverted to change the position of this 5S rDNA locus to q-arm; this metacentric chromosome thus became a submetacentric chromosome (Inv-M; Fig 3B). One more 5S rDNA locus with conspicuous FISH signals (Q1-site) was occasionally detected at subtelomeric region of the q-arm of the Inv-M chromosome (Fig 3C). A metacentric chromosome, apparently shorter than the other M-type chromosomes within same

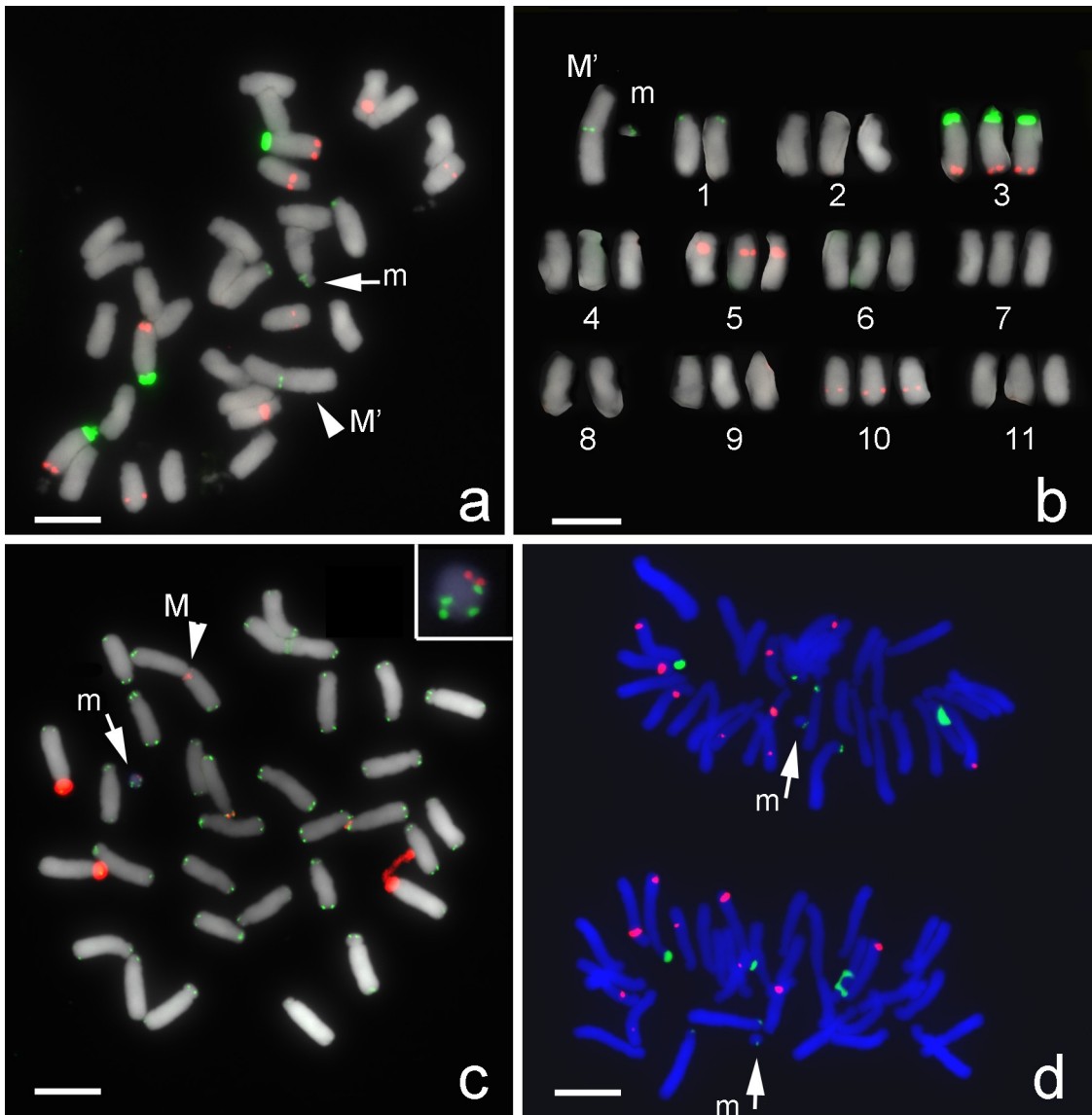

**Fig 1. Centromeric fusion detected by FISH with 45S rDNA probe in a triploid variant with karyotype 2n = 33, 31A+1M'+m.**
(Scale bar = 10 μm). (a) FISH with 45S rDNA (green) and 5S rDNA (red) as probes. FISH signals of 45S rDNA (green) were detected at the centromeric region of a large M-type chromosome (M'; arrowhead) and at a small M-type chromosome (m; arrow). (b) Individual chromosomes from Fig 1A were arranged according to their FISH signal patterns, morphology, and lengths in descending order. The M' chromosome was possibly formed by a fusion between chromosome 1 and chromosome 8. (c) FISH with 45S rDNA and telomeric DNA as probes. Both 45S rDNA (red) and telomeric sequence (green) were detected at m chromosome (arrow) and M' chromosome (arrowhead). Inset shows an enlarged image of the m chromosome with two 45S rDNA signals (red) and four telomere signals (green). (d) FISH with 5S rDNA (red) and 45S rDNA (green) as probes on chromosomes at anaphase. Two m-type chromosome (arrows) were even separated into two daughter cells.

complement, had two FISH 5S rDNA signals, one weak signal near the centromere (C-site) and one conspicuous signal at the intercalary region of the p-arm (P-site; Fig 3D). Such a shortened M-type chromosome suggested a distal segment with the 5S rDNA locus (Q-site) at the q-arm was deleted from the Inv-M chromosome. Thus, the different distribution patterns of 5S rDNA loci and arm ratios (short arm/long arm) on this M-type chromosome demonstrated the results of pericentric inversion, gene amplification, and segment deletion (Fig 3E).

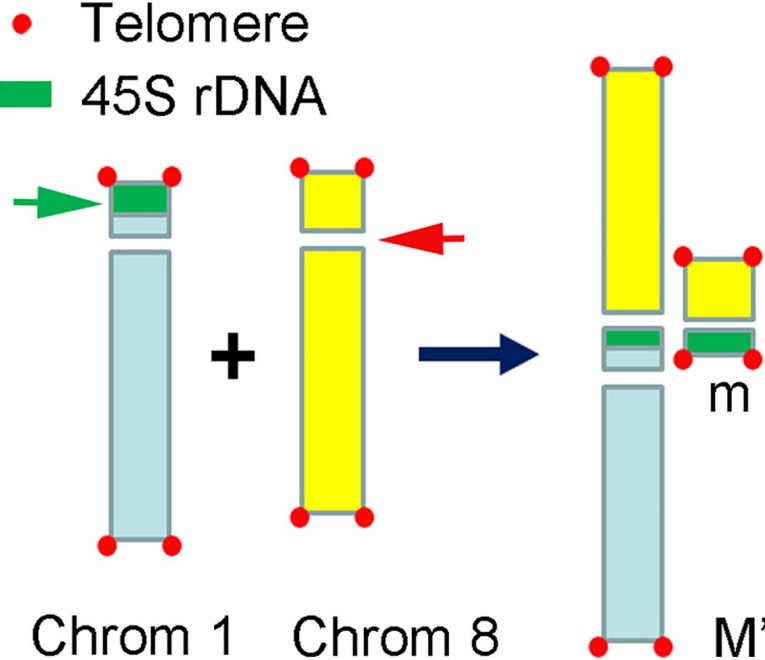

**Fig 2. Ideogram showing the occurrence of fusion between two A-type chromosomes to form the M'-type chromosome and m-type chromosome.** Chromosome fusion occurred between chromosome 1 (1) and chromosome 8 (8) after two breaks, one (green arrow) within the array of tandem repeats of 45S rDNA (green) on chromosome 1 and another (red arrow) at the centromeric region of chromosome 8.

The measurements of 15 normal and 15 inversion chromosomes showed that the centromere position (short arm/total length) changed from 0.45 to 0.32 by pericentric inversion, and the positions of three 5S rDNA-FISH signals (relative to total length) changed from P-site (0.15 ±0.01)/C-site (0.45±0.02)/Q-site (0.95±0.01) to C-site (0.32±0.03)/P-site (0.63±0.03)/Q-site (0.95±0.01) by such inversion.

Micronuclei were often observed in interphase cells of the artificial hybrid *L. aurea × L. radiata*. FISH results indicated that these micronuclei were essentially small chromosomal fragments, or mini-chromosomes, with 5S rDNA and telomeres (Fig 3F, 3H and 3I), but without 45S rDNA (Fig 3G). These mini-chromosomes could be evenly separated into daughter cells (Fig 3I) or lagged during anaphase, thus, these mini-chromosomes should have functional centromeres. At the end of mitosis, duplicated mini-chromosomes possibly remained in one of daughter cell, such uneven segregation would lead to some cells with more than one mini-chromosome (Fig 3H).

## Identification of centromeric histone 3 (CENH3) genes of *Lycoris*

The CENH3 gene in *Lycoris* was amplified from *L. aurea* cDNA by using a pair of degenerated primers designed according to the *CENH3* sequence of four *Allium* species. Among the clones of PCR products, one, named *LaCENH3* (GenBank accession no. MW036651), contained a 468-bp insert and exhibited 62.8% to 65% DNA sequence similarity to *Allium* CENH3s in the open reading frame. The DNA sequence of *LaCENH3* was predicted to encode a protein of 155 amino acids with a molecular weight 17.7 kDa. The predicted LaCENH3 exhibited more than 60% identity to *Allium* CENH3s (Fig 4). Protein analysis by using the Simple Modular Architecture Research Tool (http://smart.embl-heidelberg.de/) revealed that LaCENH3 contains a conserved H3 functional domain similar to that of canonical H3 and a remained N-

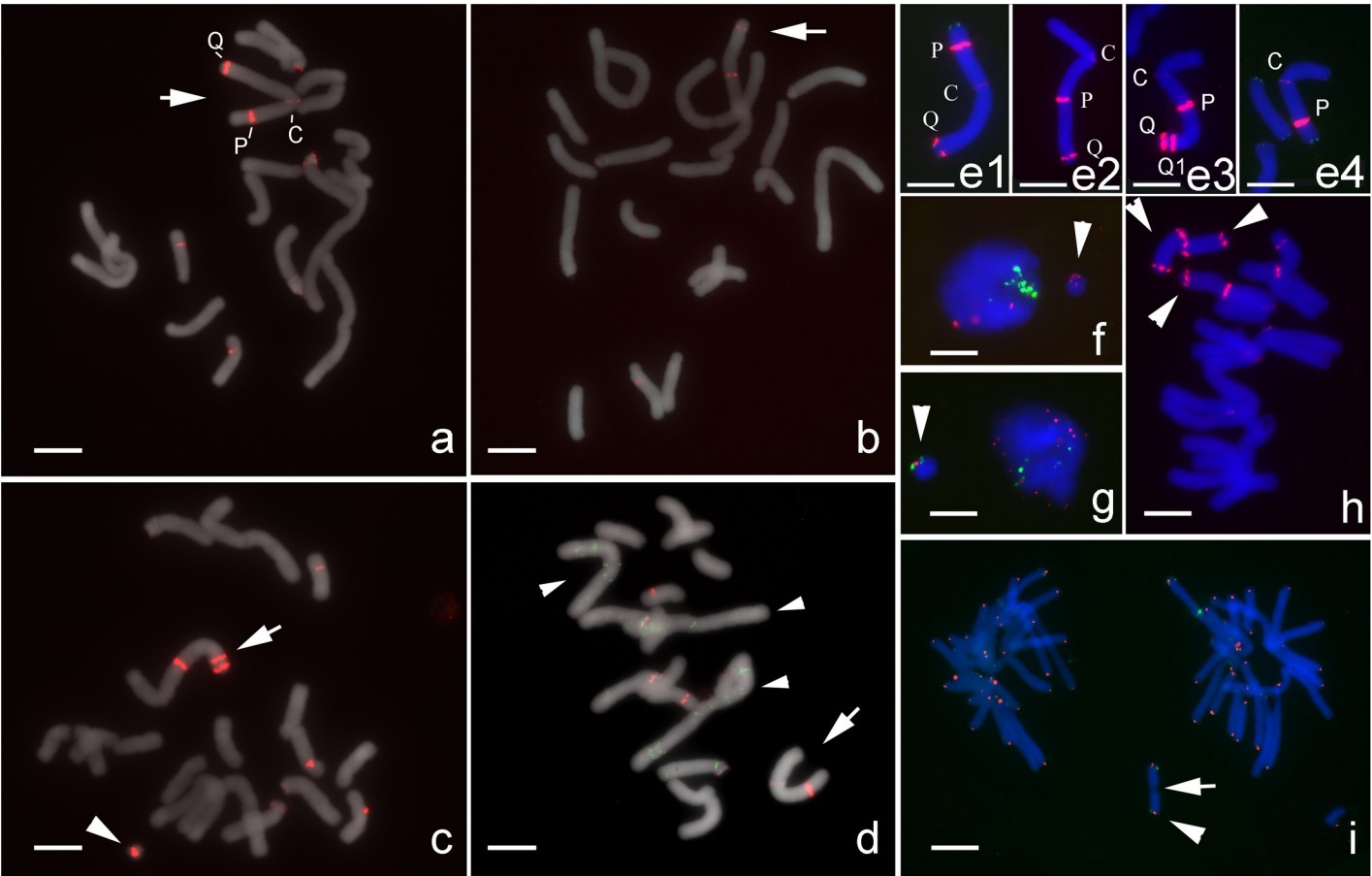

**Fig 3. FISH with 5S rDNA detected an inversion and related chromosome structural changes on one M-type chromosome in an artificial hybrid *L. aurea* × *L. radiata* (2n = 18, 4M+3T+11A).** (Scale bar = 10 μm). (a) Three FISH signals of 5S rDNA (red) were detected at one M-type chromosome (arrow), including one weak signal near the centromere (C-site) and two conspicuous signals at the intercalary region of the p-arm (P-site) and at the distal end of q-arm (Q-site). (b) Two conspicuous FISH signals of 5S rDNA (red) were detected at the q-arm and one weak signal near the centromeric region on a submetacentric chromosome (arrow). (c) One more conspicuous FISH signal of 5S rDNA (red, Q1-site) was detected near Q-site at the subtelomeric region of a submetacentric chromosome (arrow). Signals of 5S rDNA (red) were visible on the mini-chromosome (arrowhead). (d) Two FISH signals of 5S rDNA (red) were detected at an M-type chromosome (arrow), one conspicuous signal at the intercalary region and one weak signal near the centromeric region. Note that this M-type chromosome is shorter than the other three M-type chromosomes (arrowheads) in the complement. Green signals indicate the locations of LaCenIP-1 repeats which was also used as probe in this FISH experiment. (e) Different FISH signals of 5S rDNA (red) distribution patterns on one M-type chromosome (e1) show the results of a pericentric inversion (e2), amplification of 5S rDNA (e3), and deletion of a distal segment (e4). Green FISH signals in (e4) indicate the telomeres. (f-i) FISH assays characterized the structure of micronuclei in mitotic cells. (f) 5S rDNA (red), 45S rDNAs (green). No detectable 45S rDNA on micronucleus (arrowhead). (g) A micronucleus (arrow) with one 5S rDNA (red) and two telomeres (green). (h) Multiple mini-chromosomes (arrowheads) in a complement with two 5S rDNAs (red) at both distal ends. (i) A mini-chromosome (arrowhead) at anaphase with FISH signals of 5S rDNA (red) and telomere (green) possibly with a centromere (arrow).

terminal region. On protein sequence alignment, the H3 domain was conserved between LaCENH3 and *Allium* CENH3s, with almost 70% protein identity; however, the N-terminal sequences are diverse, with only 14% protein identity (Fig 4). Despite high protein identity in the H3 domain, the subdomains showed diverse protein identities: αN-helix (90% identity), α-1-helix (54.5% identity), Loop 1 (55.6% identity), α2-helix (62.1% identity), Loop 2 (83.3% identity) and α3-helix (100% identity).

## Immunodetection of LaCENH3

The efficiency and specificity of an antibody against A1-S20 at the N-terminal of LaCENH3 was confirmed by western blot analysis of nuclear proteins and the immunoprecipitated

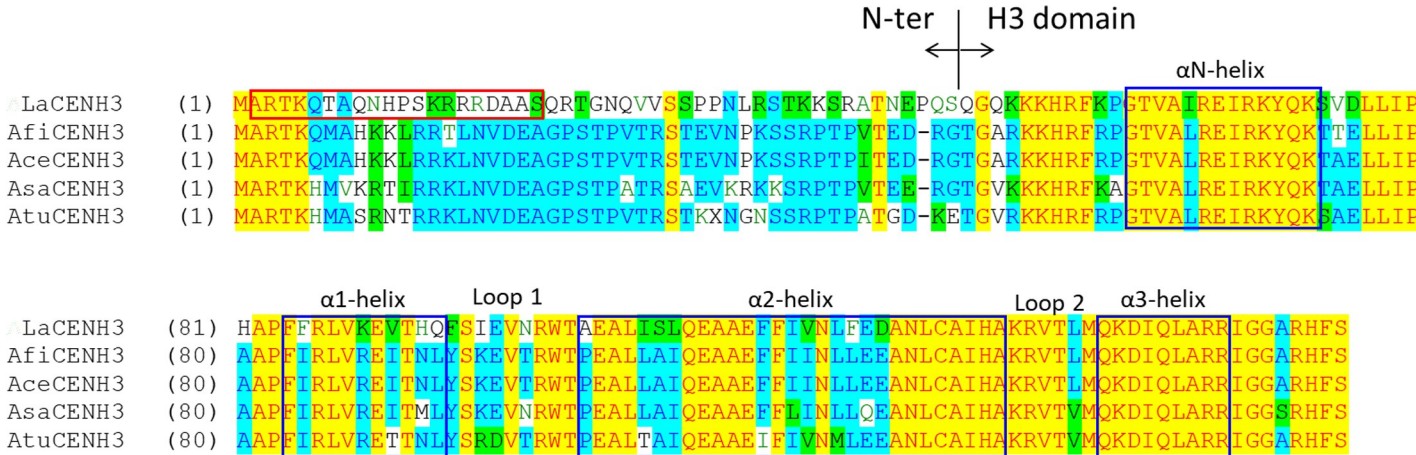

**Fig 4. Multiple amino acid sequence alignment of CENH3s of *Lycoris aurea* (LaCENH3) and *Allium* species including *A. fistulosum* (AfiCENH3), *A. cepa* (AceCENH3), *A. sativum* (AsaCENH3), and *A. tuberosum* (AtuCENH3).** Identical amino acids are indicated by the same colors. The left side of the vertical bar is the N-terminal region (N-ter) and the right side is the histone fold domain (H3 domain). The N-terminal is less conserved as compared with the H3 domain. Six functional domains are indicated in blue boxes. The sequence of peptide used to produce the anti-CENH3 antibody is indicated in the red box.

proteins with the anti-LaCENH3 antibody (S1 Fig). Cytological immunostaining with anti-LaCENH3 antibody presented strong signals in the interphase nucleus (Fig 5) and indicated the centromere positions of mitotic chromosomes of both *L. aurea* and *L. radiata* (Figs 6 and 7). The immuno-signals on the interphase nucleus appeared as spots of various size and intensity, which indicated different abundance of LaCENH3 protein at individual centromeres (Fig 5). The number of signals differed among nuclei because some centromeres embedded within

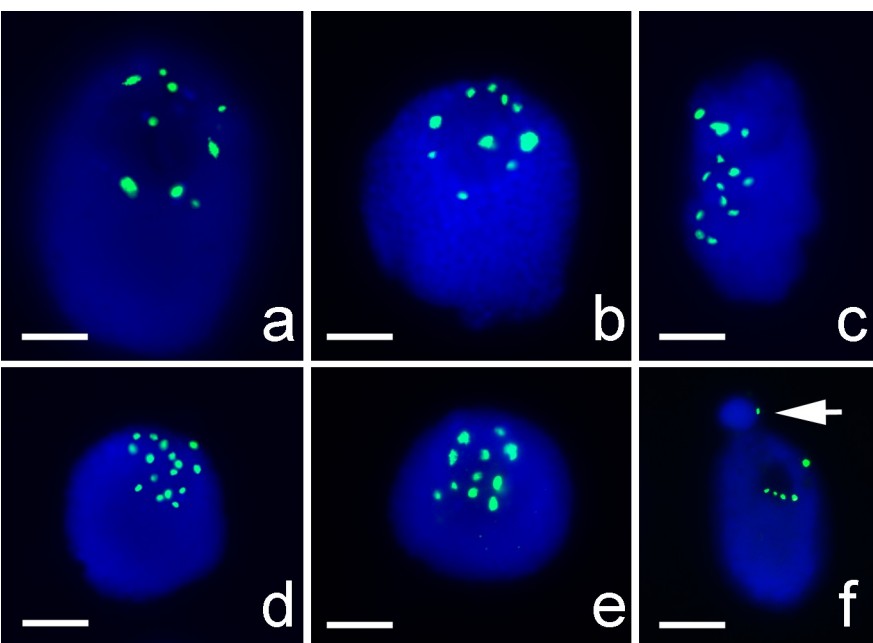

**Fig 5. Immunostaining of interphase nuclei of an artificial hybrid *L. aurea* × *L. radiata* (2n = 18, 4M+3T+11A) using an anti-LaCENH3 antibody.** (Scale bar = 10 μm). Immunostaining signals (green) are visible in interphase nucleus (blue) of root-tip meristem. Fluorescent signals in various size and intensities indicate different abundance of LaCENH3 proteins at individual centromeres. Arrow points to a micronucleus with immunostaining signals.

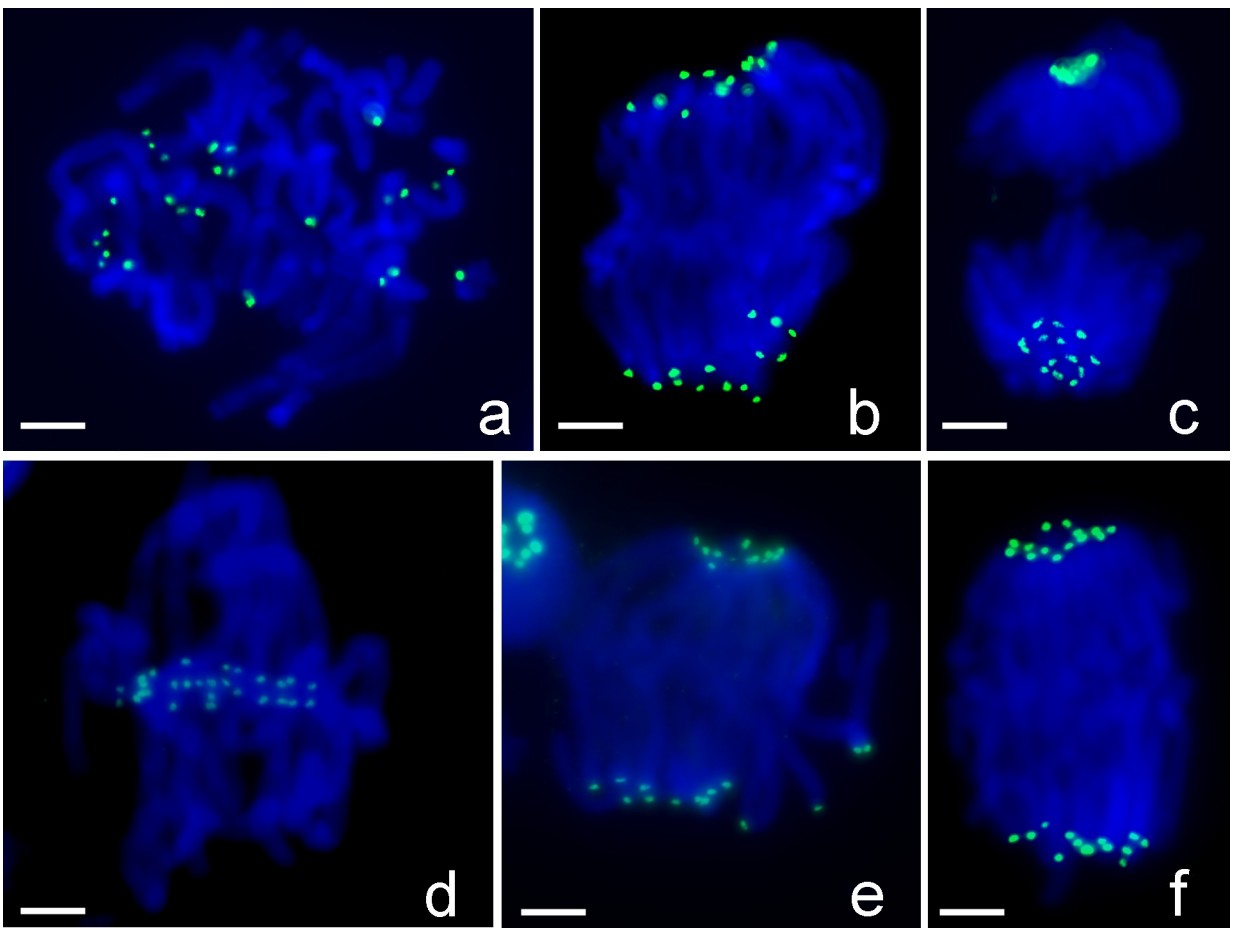

**Fig 6. Immunostaining with anti-LaCENH3 antibody visualizing centromeres on chromosomes at mitotic metaphase and anaphase.** (Scale bar = 10 μm). Immunofluorescence signals (green) indicate the centromere locations of chromosomes (blue). (a-d) Artificial hybrid *L. aurea* × *L. radiata* (2n = 18, 4M+3T+11A), (e-f) *L. radiate* (2n = 22A); (a, d) Metaphase, (b, e, f) Anaphase, (c) Telophase.

huge nuclei were difficult to label. Immuno-signals of anti-LaCENH3 antibody indicated the locations of centromeres while the chromosomes align along the metaphase plate of the spindle apparatus (Fig 6A and 6D). The distribution of immunostaining signals showed that the centromeres of sister chromatids of each chromosome pulled apart and moved to the opposite ends of the cell at anaphase (Fig 6B, 6C, 6E and 6F). The immune-signals of anti-LaCENH3 antibody at the centromere of individual chromosomes confirmed that the LaCENH3 protein, a CenH3 variant, is specific to centromeres of M-, T- and A-type chromosomes of *Lycoris* species (Fig 7). The immunostaining signals at a submetacentric chromosome indicated that was the inversion M-type chromosome with centromere position changed, and the immunostaining signal on micronuclei (mini-chromosomes) suggested the existence of functional centromeres (Fig 7). Two immune-signals on one T-type chromosome, one at the distal end of short arm and another at the proximal region of the long arm, indicated it was a dicentric chromosome with two active centromeres (Fig 7).

## Identification of centromere-related sequences

Among 10,000 ChIP clones, about 500 clones showing abundance in *L. aurea* genome by dot blot analysis were selected for further sequencing and FISH analyses. Most of these clones

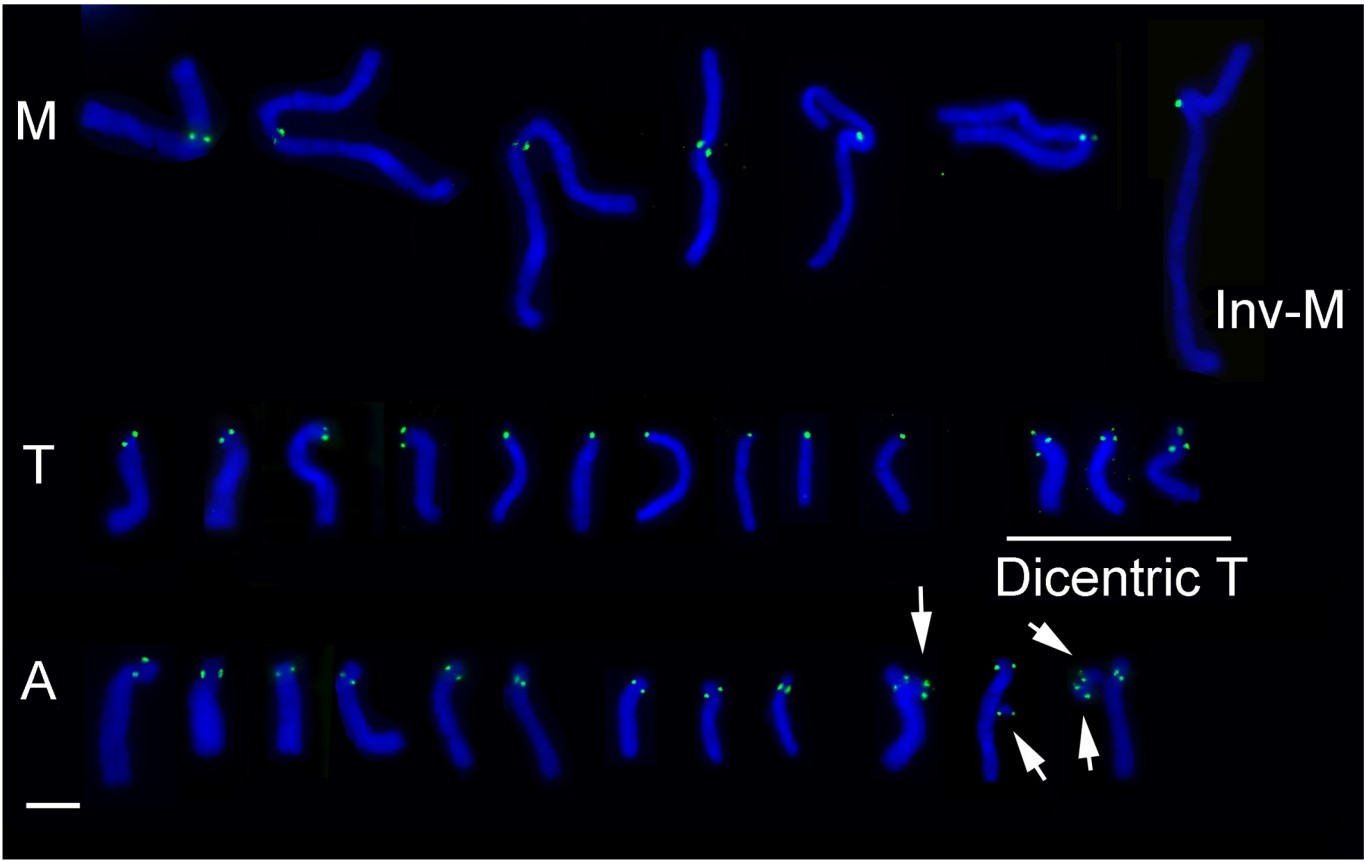

**Fig 7. Immunoassay of LaCENH3 showing specificity at centromeres of *Lycoris* chromosomes.** (Scale bar = 10 μm). Immunofluorescence signals (green) are specifically present at the centromeres of M-type, T-type, and A-type chromosomes. Note that the centromere position changed on Inv-M due to pericentric inversion on an M-type chromosome. Immuno-signals on the micronuclei (arrows), which were coincidently and random near these A-type chromosomes, confirm the presence of functional centromeres. Two green signals on some T-type chromosomes, one at the distal end of short arm (arrowhead) and one at the proximal end of the long arm (double arrowheads), suggest the existence of two functional centromeres (dicentric-T).

produced strong signals dispersed on M- and T-type chromosomes but less signals on A-type chromosomes (Fig 8A). Only five ChIP clones—LaCenIP-1, LaCenIP-2, LaCenIP-3, LaCenIP-4, and LaCenIP-1R-E1—could generate bright FISH signals at the centromeric regions of several chromosomes, which implied their association with centromeric DNA sequences (Figs 8B and 9). The term "centromeric region" is used here to include both centromeric and pericentromeric regions because these are difficult to distinguish cytologically on *Lycoris* chromosomes. For example, FISH signals of the probe LaCenIP-1 presented at the centromeric regions of four M-type chromosomes at different intensities, which indicated different numbers of LaCenIP-1 repeated at individual centromeric regions (Fig 8B). On T-type chromosomes, distinct FISH signals presented at the proximal end of the long arm but not the centromeres at the distal end of the short arm as indicated by immunostaining (Fig 7). In addition, FISH analyses using these ChIP clones generated some pale, fuzzy hybridization signals that dispersed along whole M- and T-type chromosomes but less on A-type chromosomes (Figs 8 and 9). The distribution of these five ChIP-DNAs on chromosomes were highly similar but not identical, as indicated by co-hybridization with LaCenIP-1 and three other ChIP DNAs on FISH (Fig 9). Both LaCenIP-1 and LaCenIP-2 were detected on one micronucleus (Fig 9B).

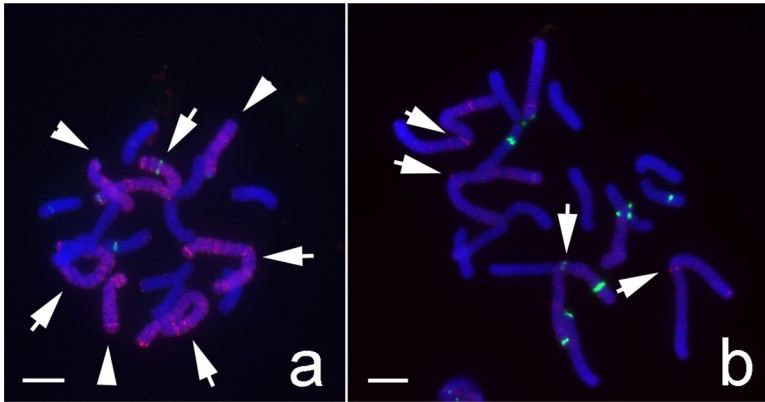

**Fig 8. FISH assay showing the abundance and specific distribution of LaCENH3-associated ChIP DNAs on mitotic metaphase chromosomes of hybrid *L. aurea* × *L. radiata* (2n = 18, 4M+3T+11A).** (Scale bar = 10 μm). (a) An example of FISH signal pattern for most of the ChIP clones tested shows strong signals (red) dispersed on M-type (arrow) and T-type (arrowhead) chromosomes, but less signals on A-type chromosomes. The green signals indicate the locations of 5S rDNA on individual chromosomes. (b) The FISH signals of LaCenIP-1 (red) show specific distribution pattern on chromosomes. Note the bright FISH signals at the centromeric regions of M-type chromosomes (arrow). The green signals indicate the locations of 5S rDNA on individual chromosomes.

Among the five centromere-related sequence clones, FISH analysis with the LaCenIP-1R-E1 probe generated the strongest and most specific signals at centromeric regions of M-type chromosomes and fewer pale signals dispersed on whole chromosomes than with the other probes (Figs 10 and 11). Furthermore, FISH with the LaCenIP-1 and LaCenIP-1R-E1

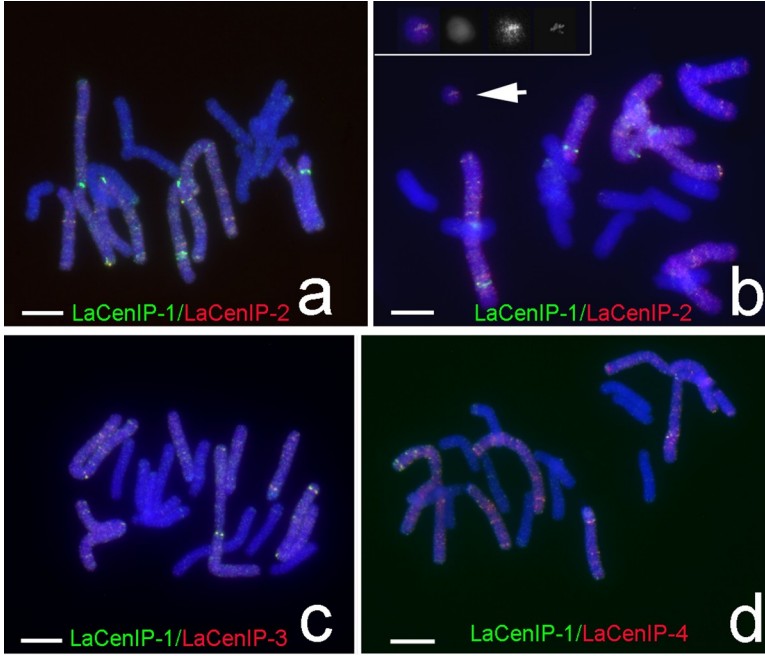

**Fig 9. FISH signals showing the similarity and different distribution patterns of ChIP-clones on *Lycoris* chromosomes.** (Scale bar = 10 μm). (a, b) LaCenIP-1 (green), LaCenIP-2 (red); (c) LaCenIP-1 (green), LaCenIP-3 (red); and (d) LaCenIP-1 (green), LaCenIP-4 (red). The FISH signals in (b) were over enhanced to show signals on the mini-chromosome (arrow). Inset shows enlarged images of that mini-chromosome in each channel: from left to right are merged image, blue signal (DAPI), red signal (LaCenIP-2) and green signal (LaCenIP-1).

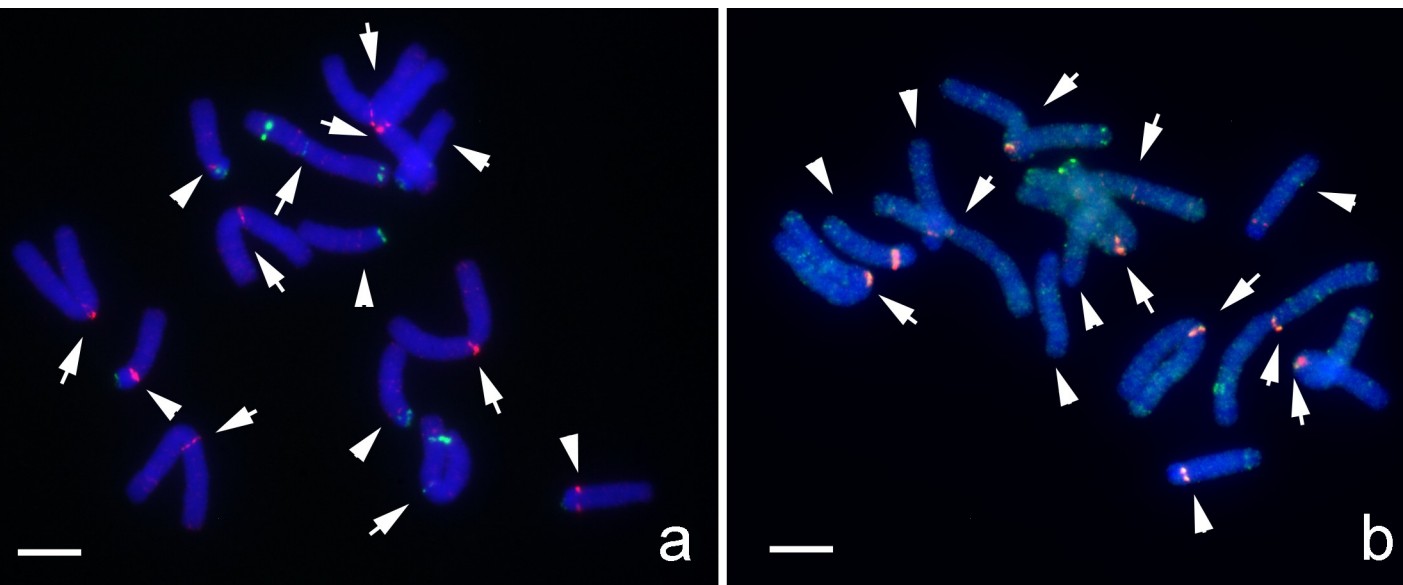

**Fig 10. FISH mapping LaCenIP-1R-E1 specifically at the centromeric region on M-type chromosomes of *L. aurea*.** (Scale bar = 10 μm). The fluorescent signals of LaCenIP-1R-E1 probes are more specifically present at centromeric regions of M-type chromosomes (arrows) in comparison with the LaCenIP-1 probe. Note the major signals of both probes are not present at distal end of T-type chromosomes (arrowheads). (a) 5S rDNA (green), LaCenIP-1R-E1 (red); (b) LaCenIP-1R-E1 (red), LaCenIP-1 (green).

probes showed unique distribution patterns on each chromosome, which allowed for identifying individual chromosomes (Fig 11 and S2 Fig).

## *Lycoris* centromere-related sequences

The sequences of ChIP clones that could generate FISH signals at centromeric regions of *Lycoris* chromosomes were deposited in GenBank with the accession numbers as the following: LaCenIP-1 (MW036637), LaCenIP-2 (MW036638), LaCenIP-3 (MW036639), LaCenIP-4 (MW036640), and LaCenIP-1R-E1 (MW036645). These five centromere-related sequence clones, LaCenIP-1, LaCenIP-2, LaCenIP-3, LaCenIP-4, and LaCenIP-1R-E1 were 590, 607, 517, 524, and 778 bp long, respectively. No DNA sequences of the NCBI database (https://blast.ncbi.nlm.nih.gov/Blast.cgi) were found homologous to these ChIPed DNAs by blastn analysis. Among the five sequences, LaCenIP-1, LaCenIP-2, LaCenIP-3, and LaCenIP-4 showed a higher similarity to each other. Sequence alignment revealed that LaCenIP-1, LaCenIP-2, LaCenIP-3, and LaCenIP-4 shared a 374-bp overlap region. Pair-to-pair comparison indicated high sequence identity at the respective overlap regions (Table 1). The DNA sequences for the 374-bp overlap region exhibited 55.9% identity among these four clones, with identity 84.8% on excluding LaCenIP-3 from the comparison. The sequence of LaCenIP-3 seemed less identical to the other three sequences. However, the sequence of LaCenIP-1R-E1 (778 bp) exhibited 41.8% to 44.3% identity to LaCenIP-1-4 in the full-length sequence but 55% to 59.1% identity when considering only the overlap region.

## Discussion

### Chromosomal variants in *Lycoris*

Extensive cytology study has revealed a great variability in chromosome number and karyotype among *Lycoris* species [2]. Several chromosomal variants such as M', T', m and a, despite

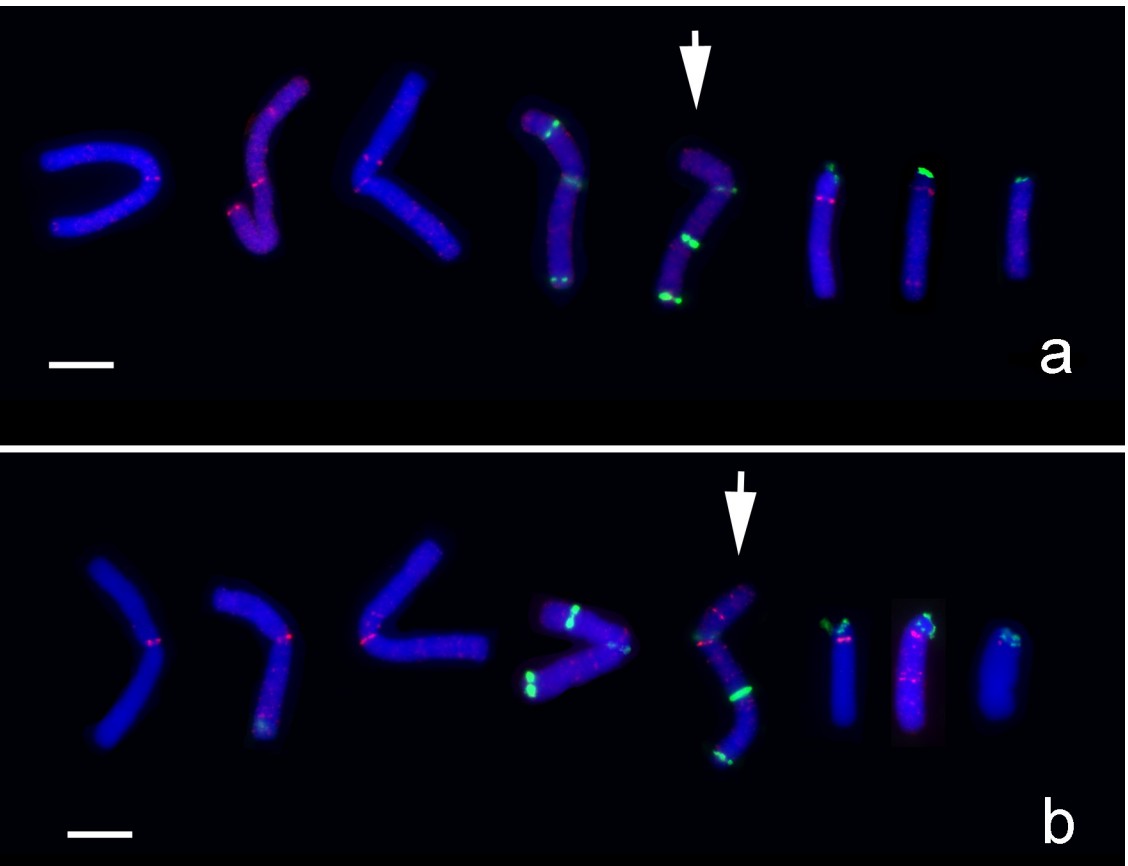

**Fig 11. Karyotype of mitotic metaphase chromosomes of *L. aurea* with distribution of LaCenIP-1 and LaCenIP-1R-E1 repeats along chromosomes to distinguish individual chromosomes.** (Scale bar = 10 μm). Karyotype is established by individual chromosomes with clear and represented FISH signal distribution patterns. Note the similarity and difference of FISH patterns between probes on each chromosome. (a) LaCenIP-1 (red), 5S rDNA (green); (b) LaCenIP-1R-E1 (red), 5S rDNA (green). Arrow indicate the inversion M-type chromosome (arrow).

low frequencies, have been observed in wild populations of *Lycoris* species. The variants were considered products of various structural changes including translocation, deletion, and inversion on three fundamental chromosome types [36]. Despite the large size of the chromosomes, the interphase nuclei and mitotic chromosomes presented less distinctively condensed blocks of heterochromatin and showed a diffused pattern by aceto-orcein staining and C-banding [2]. Therefore, deducing the chromosomal rearrangements in the *Lycoris* genus is challenging on the basis of observations without chromosome-specific landmarks. As we reported previously [34], FISH revealed that DAPI-positive bands at the end of the p-arm of T-type chromosomes

**Table 1. Sequence identity of five centromere-related sequences by pair-to-pair comparison of their overlap regions and cloned full-length sequences (parentheses).**

| Sequence ID | LaCenIP-1 | LaCenIP-2 | LaCenIP-3 | LaCenIP-4 | LaCenIP-1R-E1 |
|---|---|---|---|---|---|
| LaCenIP-1 | - | 87.9% (45.5%) | 60.8% (50.9%) | 89.7% (81.6%) | 55% (41.8%) |
| LaCenIP-2 | 87.9% (45.5%) | - | 63.1% (42.7%) | 91.4% (42.9%) | 59.1% (44.3%) |
| LaCenIP-3 | 60.8% (50.9%) | 63.1% (42.7%) | - | 58.7% (44.2%) | 55.5% (43.2%) |
| LaCenIP-4 | 89.7% (81.6%) | 91.4% (42.9%) | 58.7% (44.2%) | - | 56.8% (44.2%) |
| LaCenIP-1R-E1 | 55% (41.8%) | 59.1% (44.3%) | 55.5% (43.2%) | 56.8% (44.2%) | - |

had highly repetitive sequences, including 45S rDNA, 5S rDNA and telomeric DNA. FISH also revealed a DAPI-positive band near the centromeric region of one M-type chromosome.

In this study, more chromosomal variants were visualized by FISH with using 45S rDNA, 5S rDNA and telomeric sequences. FISH signal patterns on chromosomes of a triploid (2n = 33, 31A+M'+m) indicated that the large M-type chromosome (M') and small M-type chromosome (m) were produced by centric fusion of two A-type chromosomes (Figs 1 and 2). Also, FISH signals revealed the occurrence of a pericentric inversion and sequential amplification of 5S rDNA and segment deletion on an M-type chromosome (Fig 3). The ChIP clones we obtained could generate distinct FISH patterns on each chromosome (Fig 11), which were also useful markers for identifying individual M- and T-type chromosomes.

Plants with karyotype 2n = 33, 31A+1M'+m have been identified in several populations with standard karyotype 2n = 33A [37, 38]. In several *Lycoris* species, in some cases, the small chromosomes were considered B or supernumerary chromosome [2, 4, 35, 38–40]. As we showed here, the m chromosome with 45S rDNA and telomeres (Fig 1A–1C) was evenly segregated to daughter cells in anaphase (Fig 1D) and so considered as an intact chromosome with a functional centromere.

Robertsonian translocation is known as an important process in karyotype evolution leading to speciation in *Lycoris*. Much evidence supported both the fusion and fission translocation response to diverse karyotypes among *Lycoris* species, including cytological observations [6, 36, 41–44] and phylogenetic analysis based on the chloroplast genome [45]. Previously, we suggested that the T-type chromosome was formed by centric fission of M-type chromosomes because of a large block of 45S rDNA and 5S rDNA adjacent to the telomeric repeats detected at the physical end by FISH [34]. The existence of abundant repeats demonstrated amplification of these repeats to cap the broken ends [34]. Here, results of FISH with 45S rDNA probe showed fusion of two A-type chromosomes leading to form one large M-type chromosome (M') and one small M-type chromosome (m) in a plant found among population with 2n = 33A karyotype (Fig 1). Robertsonian fusion must involve two breaks on reciprocal chromosomes; in this case, two breaks were found within large blocks of repetitive DNA sequences, one in a cluster of 45S rDNA repeats and another in the centromeric region (Fig 2).

Interspecific hybridization and subsequent polyploidy in natural habitats were considered important in speciation within the genus *Lycoris* [1, 3, 46–48]. The merging of two divergent genomes in a hybrid is believed to trigger a genomic shock [49], which would alter gene expression and epigenetic control of transposable elements, thus leading to genomic rearrangements, changes in genome size, chromosomal rearrangement, and somatic variations [50]. Therefore, the chromosomal structure changes we observed here (Fig 3) as well as other chromosomal variants discovered in putative hybrid taxa of *Lycoris* [36, 37] suggested a common occurrence of chromosome structural changes in interspecific hybrids.

These aberrant chromosomes could be retained in subsequent vegetative generations because *Lycoris* plants can propagate without seeds, and the most important is the variant chromosomes with functional centromeres for normal segregation at anaphase and being inherited by daughter cells. Our results of FISH and immunostaining showed that these mini-chromosomes were intact chromosomes with 5S rDNA and telomeres (Fig 3F, 3G and 3I) and possessed active centromeres for normal segregation at anaphase (Fig 3I), which suggested the formation of new centromeres on these mini-chromosomes in response to the genomic instability of interspecific hybrids.

## Centromere-related DNA sequences in *Lycoris*

Plant centromeres consist of multiple satellite repeats that are highly diverse even among closely related species [9, 51, 52]. The length of monomer centromeric satellite DNA ranges from 50 to 2094 bp [24] and most are 150–180 bp [9]. The centromeric satellite repeats of related species are highly diverse in nucleotide composition and genomic abundance. In cereal centromeres, the centromere-specific retrotransposons (CRs) were mostly Ty3/gypsy type retrotransposons [53] and other retrotransposons [54, 55].

The centromeres of rice (*Oryza sativa*, AA genome) consist of tandem arrays of CentO repeats (155 bp) and interspersed rice-specific CRs (CRRs) [17, 56, 57]. However, some wild *Oryza* species contain other CRRs, for example, CentO-F (156 bp) in centromeres of *O. brachyantha* (FF genome), which share no homology with the CentO repeats or other centromeric DNA in grass species [58]. In most studied species, the centromere-specific repetitive DNAs show high sequence similarity among the centromeres within a complement, as in *Arabidopsis thaliana* [53], *O. sativa* [56], *O. brachyantha* [58], and maize [54]. However, centromeres on individual chromosomes within a complement had unique and multiple repeats as reported in potato [59], pea [24], and bean [25]. Each centromere of cultivated potato (*Solanum tuberosum*, 2n = 4x = 48) contains distinct DNA sequences in different organizations [59]. In pea (*Pisum sativum* L. 2n = 14), each chromosome has a very large single centromere that contains multiple functional domains associated with clusters of distinct satellite DNA families [24]. In bean (*Phaseolus vulgaris* L. 2n = 2x = 22), two unrelated centromere-specific satellite repeats, CentPv1 (99-bp) and CentPv2 (110-bp) were identified and mapped at two subsets of centromeres [25]. In this study, an antibody against the centromere-specific H3 variant of *L. aurea* could label all centromeres of both *L. aurea* (2n = 14, 8M+6T) and *L. radiata* (2n = 22, 22A; Figs 5–7) and was used in ChIP to precipitate CENH3-associated DNA fragments. FISH results showed five of 500 ChIP clones located specifically on the centromeric region of M-type chromosomes (Figs 8–11). On T-type chromosomes, the FISH signals of these five ChIP clones were mainly clustered at the proximal region of the long arm (Figs 8–11) but not the centromere at the distal end of the short arm as the immunostaining signals indicated (Fig 7). No clusters of FISH signals were detected on A-type chromosomes. In addition to clusters of blight signals, there were more or less pale, fuzzy hybridization signals dispersed along whole M- and T-type chromosomes but fewer signals on A-type chromosomes (Figs 8 and 9). Signals for some centromere-specific retrotransposons (CRs) were also found in non-centromeric regions, such as pericentromeric and interstitial regions on chromosomes of rice, wheat, and Brassica [57, 60–62].

These five ChIP-clones with relative similarity in DNA sequence (Table 1) showed similar but not identical distribution on chromosomes by FISH-mapping (Fig 11). Therefore, these five ChIP-cloned DNAs were the centromeric satellite repeats of M-type chromosomes but not T- and A-type chromosomes. That is, centromeres of different type chromosomes of *Lycoris* species may be associated with distinct satellite DNA families. In comparing the distribution pattern of FISH signals (Figs 8–11) to those of immunostaining signals (Fig 7), *Lycoris* centromeres contain more and diverse satellite repeats than discovered in this study. The antibody against CENH3 obtained in this study would be useful for further ChIP-cloning of more centromere-associated DNA sequences of *Lycoris* species.

The mini-chromosomes observed in the artificial hybrid *L. aurea* × *L. radiata* could be evenly separated into daughter cells or lagged during anaphase (Fig 3I) and were confirmed to associated with CENH3 by immunostaining (Figs 5F and 7). FISH results indicated that these mini-chromosomes possess 5S rDNA, telomeres (Fig 3F, 3G and 3I), and centromeric-related sequences (Fig 9B). Thus, the chromosome fragments with 5S rDNA and telomeres each have

a functional centromere. More evidence is needed to prove whether those two centromeric-related sequences are directly associated with CENH3, or de novo neocentromere formation occurred on these small chromosome fragments.

A neocentromere is a de novo site with centromere function without canonical centromeric repeats. Again, centromeric chromatin is defined by the presence of CENH3 not by specific DNA sequences, which is an epigenetic component to centromere specificity. Neocentromeres would be formed to rescue acentric chromosome fragments from being lost in subsequent segregation, as reported in barley [63], oat-maize addition lines [64], and maize [65–67]. The antibody against CENH3 we obtained would be useful to identify more centromeric repeats, which can shed light on understanding chromosome structural variation and the mechanism of neocentromere formation in *Lycoris*.

Two CENH3-associated sites were detected at one T-type chromosome by immunostaining with the CENH3 antibody, one at the distal end of the short arm and another at the proximal region of the long arm (Fig 7); only the latter presented significant FISH signals of the five centromeric-associated ChIP clones (Figs 8–11). According to the definition of an active centromere, this T-type chromosome presented two active centromeres. A chromosome with two centromeres is a dicentric chromosome, commonly found in many plants, such as maize B-A translocation chromosome derivatives [68], wheat-barley translocation lines [63], and wheat-rye translocation lines [69]. Two active centromeres per chromosome will undergo chromosome breakage during cell division [68, 70–72], with only one of the two recognized with centromeric activity. Although the DNA sequence of two centromeres on a dicentric chromosome is identical, the smaller centromere usually becomes inactive, as reported in maize [68, 73] and wheat [71]. For the dicentric T-type chromosome we observed (Fig 7), no FISH signals with ChIP clones were detected at the distal end of the short arm (Figs 8–11), conventionally recognized as the centromere of T-type chromosomes. FISH results alone were insufficient to determine whether different DNA sequences or identical DNA sequences had different abundance on these two centromeres for this T-chromosome.

The genomes of *L. aurea* (MT-karyotype) and *L. radiate* (A-karyotype) showed high homology according to genomic in situ hybridization (GISH) analysis of an interspecific hybrid of *L. aurea* × *L. radiata* [33]. GISH results showed various recombinants resulting from homoeologous recombination between the MT- and A-genome. Recombination occurred throughout these chromosomes but rarely in the pericentric regions [33]. However, GISH experiments also suggested considerable divergence between both genomes because 60-fold unlabeled blocking DNA in excess of probe DNA was added in the GISH mixture to discriminate MT- from T-type chromosomes [33]. All these investigations indicated the existence of relative affinity between the MT-genome and A-genome allowing frequently homoeologous recombination and leading to partial fertility in an interspecific hybrid of *L. aurea* × *L. radiata*. In contrast to GISH using total genomic DNA as a probe, FISH probing with ChIP-cloned centromeric DNAs in this study revealed less homology between *L. aurea* and *L. radiata* genomes. Most ChIP-cloned DNAs from *L. aurea* generated bright FISH signals evenly distributed on M- and T-type chromosomes but rarely on A-type chromosomes (Fig 8A). In addition to chromosomal structure changes, the accumulation of numerous alterations in DNA composition and genomic organization during evolution could result in genome-specific repetitive sequences, which allowed for differentiating related genomes at the chromosome level [74]. *Lycoris* species have a giant genome size: the estimated genome size is larger for *L. aurea* than *L. radiata*, 1C-value = 30.40 pg vs 20.22–25.46 pg [27]. The cytological estimation of chromosome size in a complement of the *L. aurea* × *L. radiata* hybrid also showed greater proportion of M- and T-type chromosomes than A-type chromosomes (57.03% vs 42.87%) [33]. Therefore, not unsurprisingly, the MT-genome contains abundant and diverse repetitive

DNA specifically distributed on M- and T-type chromosomes but less on A-type chromosomes.

Studies of the *Lycoris* genome are still limited because of the large genome sizes. For *Lycoris* lacking a reference genome and information on well-annotated genes, gene and ChIP cloning are faithful ways to study centromeric-related sequences. ChIP using the CENH3 antibody is a feasible way to obtain centromeric-specific repeats in many plants. However, the efficiency of ChIP could be decreased because of dispersed repetitive sequences within the centromeric chromatin fraction or the weak cross-reaction of serum with non-centromeric chromatin. Immunostaining results in this study showed high specificity of the CENH3 antibody to recognize centromeres of all chromosomes (Fig 7), but only five of more than 10,000 ChIP clones were characterized as centromeric DNAs (Figs 8–11). These results demonstrate the difficulty in isolating centromeric repeats from such a highly complex and large genome as *Lycoris*. The DNA sequences and chromosomal distribution patterns of those ChIP clones showed both similarity and diversity, so *Lycoris* centromeric repeats may include one or more short repeats embedded and interspersed by diverse repetitive sequences.

## Conclusions

By combining ChIP cloning, cytological immunostaining, and FISH studies, we demonstrate the structural variants and different organization of the centromeric region of *Lycoris* chromosomes. FISH probing with 45S and 5S rDNA revealed centric fusion, inversion, segment deletion, and gene amplification on *Lycoris* chromosomes. Immunostaining with an antibody against centromere-specific histone H3 (CENH3) of *L. aurea* (2n = 14, 8M+6T) could label the centromeres of M-, T-, and A-type chromosomes. However, FISH analyses revealed that five CENH3-associated DNAs presented different distribution patterns on chromosomes, which indicated different organization of centromeres among individual chromosomes of *Lycoris*. In addition, immunostaining revealed a dicentric T-type chromosome and a centromere on a mini-chromosome. The roles of these five ChIP clones in centromere function are unknown from current data; however, the anti-CENH3 antibody of *Lycoris* we obtained could be useful for further studying the organization of *Lycoris* centromeres and the centromeres on chromosome variants, such as dicentric and mini-chromosomes. Also, these five CENH3-associated DNAs would be useful FISH markers for identifying individual M- and T-type chromosomes.

## Supporting information

**S1 Fig. Western blot analysis of LaCENH3 and histone 3 proteins to confirm the efficiency and specificity of the anti-LaCENH3 antibody against A1-S20 at the N-terminal of LaCENH3.** Crude nuclear proteins (lanes 1, 4 and 7), LaCENH3 antibody immunoprecipitated proteins (lanes 2, 5 and 8) and H3K4me3 antibody immunoprecipitated proteins (lanes 3, 6 and 9) were separated on a polyacrylamide gel, transferred onto a PVDF membrane and detected with pre-serum, LaCENH3 and H3K4me3 antibodies, respectively. Pre-serum was used as a negative control. The immunoprecipitated proteins were obtained from equal amount of nuclear proteins. The protein loading volume was normalized to equal amount of crude nuclear proteins. The relative molecular weight of protein was indicated at the left. (PDF)

**S2 Fig. Collection of chromosomes from several FISH experiments with LaCenIP-1 and LaCenIP-1R-E1 as probes showing unique distribution patterns consistently detected on individual chromosomes.** (Scale bar = 10 μm). (PDF)

## Acknowledgments

Thanks to Prof. Chou-Tou Shii, Ms. I-Ju Chen, and Ms. Yu-Chu Chang for their assistance with sample collection.

## Author Contributions

**Data curation:** Mao-Sen Liu, Shih-Hsuan Tseng, Ting-Chu Chen.

**Formal analysis:** Mao-Sen Liu, Mei-Chu Chung.

**Funding acquisition:** Mei-Chu Chung.

**Investigation:** Mao-Sen Liu, Mei-Chu Chung.

**Methodology:** Mao-Sen Liu, Shih-Hsuan Tseng, Ching-Chi Tsai, Ting-Chu Chen, Mei-Chu Chung.

**Project administration:** Mei-Chu Chung.

**Supervision:** Mei-Chu Chung.

**Writing – original draft:** Mao-Sen Liu, Mei-Chu Chung.

**Writing – review & editing:** Mao-Sen Liu, Shih-Hsuan Tseng, Mei-Chu Chung.

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
