## [Decision Letter · Decision Letter 0]

20 Jul 2021

PONE-D-21-19380

Chromosomal variations of Lycoris species revealed by FISH with rDNAs and centromeric histone H3 variant associated DNAs

PLOS ONE

Dear Dr. Chung,

Thank you for submitting your manuscript to PLOS ONE. After careful consideration, we feel that it has merit but does not fully meet PLOS ONE’s publication criteria as it currently stands. Therefore, we invite you to submit a revised version of the manuscript that addresses the points raised during the review process.

We look forward to receiving your revised manuscript.

Kind regards,

Zhukuan Cheng

Academic Editor

PLOS ONE

Journal Requirements:

[This work was funded by the Institute of Plant and Microbial Biology, Academia Sinica, Taiwan, ROC, and Ministry of Science and Technology, Taiwan, ROC (grant no. MOST 106-2313-B-001-007 -MY3).

The funders had no role in study design, data collection and analysis, decision to publish, or preparation of the manuscript.].    

We note that one or more of the authors are employed by a commercial company: CH Biotech R & D Co., LTD.

Reviewers' comments:

Reviewer's Responses to Questions

**Comments to the Author**

1. Is the manuscript technically sound, and do the data support the conclusions?

Reviewer #1: Yes

Reviewer #2: Yes

2. Has the statistical analysis been performed appropriately and rigorously? 

Reviewer #1: N/A

Reviewer #2: N/A

3. Have the authors made all data underlying the findings in their manuscript fully available?

Reviewer #1: Yes

Reviewer #2: Yes

4. Is the manuscript presented in an intelligible fashion and written in standard English?

Reviewer #1: Yes

Reviewer #2: Yes

5. Review Comments to the Author

Reviewer #1: 1. The writing of manuscript is poor, and also contains some typos/grammar issues. In order to improve your manuscript, the entire paper needs a thourough professional linguistic revision.

2. Please see the attached file. My comments were made directly on the MS.

Reviewer #2: Characterization of chromosome constitution and structural variations among species or accession in a specific genus is a basic study for understanding its origin, evolution and diversification. In this study, chromosome complements of Lycoris species were studied by FISH using rDNA probes and ChIPed centromeric repetitive DNA sequences, and immunostaining using a developed CENH3 antibody. They observed in addition to three fundamental metacentric (M-), telocentric (T-), and acrocentric (A-) type chromosomes,chromosomes in various morphology and size were also present in natural populations. They successfully detected several chromosomal structure changes in Lycoris including centric fusion, inversion, gene amplification, and segment deletion.Furthermore, the distinct distribution patterns of FISH signals on each chromosome generated by five ChIP clones allow to identify individual chromosome, which is considered difficult by conventional staining approaches. their results provide cytological evidences for a different organization of centromeres of the three chromosome types in Lycoris species. The figures presented are in high quality and convincing to draw the conclusion. The obtained results provide useful tools for chromosome identification and valuable information for the genome constitution of Lycoris species.

The munuscript can be accepted for publication after minor revision.

6. PLOS authors have the option to publish the peer review history of their article (what does this mean?). If published, this will include your full peer review and any attached files.

Reviewer #1: **Yes: **Yonghua Han

Reviewer #2: **Yes: **Xiue Wang

---

## [Author Response · Author response to Decision Letter 0]

29 Aug 2021

PONE-D-21-19380

Dear Academic Editor, Professor Zhukuan Cheng,

Thank you for considering the manuscript entitled " Chromosomal variations of Lycoris species revealed by FISH with rDNAs and centromeric histone H3 variant associated DNAs " (PONE-D-21-19380). We deeply appreciate all the valuable comments and constructive suggestions from you and reviewers. 

We have substantially revised the manuscript according to your comments and suggestions. We have asked a native English speaker and two colleagues to improve the English editing of the revised manuscript. The style and format of the revised manuscript has been checked according to the formatting guild lines to meet PLOS ONE’s requirements. 

Please check our responses to your suggestions and comments at the end of this letter. We have made point-by-point responses to the comments and suggestions which were noted directly on manuscript by reviewers. Please see our responses in the attachment PDF and rebuttal letter_Response to Reviewers.

We would like to submit the revised manuscript to the "PLOS ONE", and hope it is acceptable for publication in the journal. 

Again, we deeply appreciate your assistance and the contributive comments. Please do not hesitate to contact us for any question or concern.

We look forward to your final decision.

Responses to editor:

1 and 2. The issue about one of the authors (Ching-Chi Tsai) is employed by a commercial company (CH Biotech R & D Co., LTD).

Reply: This author is currently employed by CH Biotech R & D Co., LTD(CHB); but, before she moved to CHB, she worked as a post-doctor and joined this study which was supported by the funding from the Institute of Plant and Microbial Biology, Academia Sinica, Taiwan, ROC. Therefore, that commercial company, CH Biotech R & D Co., LTD, did not play any role in this study. We have revised the author’s affiliation to “Institute of Plant and Microbial Biology, Academia Sinica, Taipei, Taiwan”, and changed “CH Biotech R & D Co., LTD “as her current address.

3. The issue about “data not shown”.

Reply: We have provided the image of western blot analysis as Supporting information (S1_Fig.) and change the sentence in text (Lines 310-311 in 'Revised Manuscript with Track Changes'). 

4. The issue about sharing laboratory protocols in protocols.io

Reply: It is a good idea, we will consider preparing a manuscript to share our experience of studying centromere of plants having such a large genome size.

Responses to reviewers:

We deeply appreciate all the valuable comments and constructive suggestions from reviewers. We have substantially revised the manuscript according to your comments and suggestions. 

Responses to reviewer 1:

1. We have asked a native English speaker and two colleagues to improve the English editing of revised manuscript. 

2. We made point-by-point responses to the comments and suggestions which were noted directly on manuscript by reviewers. Please see our answers at the end of this letter and the attachment file.

Responses to reviewer #1: 

Page number, reviewer’s point marked on origin manuscript PDF: Our response (Line number in 'Revised Manuscript with Track Changes')

Page 9, Introduction: It was an editing error, “and” has been deleted (Line 62). 

Page 9, Introduction: It was a typo, “chromosomes” has been corrected to “chromosome” (Line 64). 

Page 16, Results: I have checked the reference [33], and found that the plant materials L. radiata var. radiata (2n= 3x = 33A) was not used in this study.

Response: Sorry, it should be in the reference [34]. 

Yes. L. radiata var. radiata (2n = 3x = 33, 33A) was not used in that study [Chang et al., 2009, reference #34]. However, the rDNA-FISH signals pattern shown here is identical to that on A-type chromosomes of Lycoris albiflora (2n = 17, 5M + 1T + 11A) (Fig. 5A in reference #34). Lycoris albiflora is considered a hybrid between L. traubii (2n = 12, 10M + 2T) and L. radiata var. radiata (2n = 3x = 33, 33A) (Kurita, 1987, ref #35). 

We revised this sentence (Lines 236-241) and rearranged the numbering order of references (# 34-#39, Lines 860-879). The positions of these references in text were also checked and corrected. 

Page 16, Results: Confirm it with your anti-LaCENH3 antibody.

Response: Thank you for your suggestion, we will try it out in the future; however, we delete such surmise here before we get further results. (Lines 248-249)

Page 17, Results: Please provide the picture that showed one M-type chromosomes of L. aurea (n=4M+3T) has three 5S rDNA loci in reference [33].

Response: Sorry, it should be in reference #34. The Fig. 2 in reference # 34 showed the rDNA-FISH signals on chromosomes of L. aurea. The 4 M-type and 3 T-type chromosomes of this hybrid used here were from L. aurea. (Line 254)

Page 11, Results: I can only see one FISH signals of 5S rDNA from Fig.3b.

Response: Two signals are visible in high resolution image. Please click the link at the top of each preview page to download a high-resolution version of each figure. 

Page 21: 

Response: It was a typo, we corrected it, and it is (Fig 9b) in revised manuscript (Line 354)

Page 21 and 22: Two sentences are repeated in this paragraph.

Response: The sentence in page 21 has been deleted (Lines 377-378)

Page 24, Give reference.

Response: Reference [34] has been inserted at the end of this sentence. (Line 402)

Page 26, These sentences can be deleted.

Response: We want to keep these sentences because signals of these ChIP clones in this study were also detected in non-centromeric regions. (Lines 493-495) 

Page 28, where? I can't see it from the Fig.7. 

Response: Two signals have been indicated by arrows.

Page 32. Figure legends: The location and format of “Scale bar = 10 µm” should be consistent for all figures. 

Response: The format and locations for “Scale bar = 10 µm” in all figure legends have been regulated. 

Page 32, Fig 1d: The image given does not show that the m-type chromosome was lagged between two daughter cells. 

Response: Among the observed chromosomes at anaphase, the m-type chromosomes were either lagged or even separated into two daughter cells (as shown in this image). This sentence has been revised. (Lines 659-661) 

Page 32, Figure 3: The title does not exactly match the content of all the images in Fig. 3. 

Response: The title of Fig 3 has been modified (Lines 668-672)

Page 33, Fig. 3b: I can only see one FISH signals of 5S rDNA (red).

Response: There are indeed two signals, please check the high-resolution image.

Page 33, Fig. 3c: micronucleus or mini-chromosomes?

Response: We indicated it as "mini chromosome" here. (Line 682)

Page33, Fig. 3d: The purpose of locating the clone is not stated, and the clone name is not given in the title of Figure 3.

Response: The FISH signals of 5S rDNA are helpful for identifying Lycoris chromosomes. We used 5S rDNA as reference probe with each ChIP clone in FISH analysis to screen potentially centromeric sequence. So, we have chances to find chromosomal variants from mass FISH images obtained in these experiments. Such a short M-type chromosome was occasionally observed. This cell with clear and represented 5S rDNA-FISH signals on each chromosome was observed in the FISH experiment with 5S rDNA and LaCenIP-1 as probes. However, we do not want to lay emphasis on this clone here. This sentence is changed to "Green signals indicate the locations of LaCenIP-1". (Lines 687-690)

Page 33, Fig. 3e: It would be better to replace pictures a-d with complete cells with chromosomes e1-e4.

Response: Each of pictures Fig 3a-d shows a complete cell with each different type of chromosomal variant. Chromosomes e1-e4 were arrayed here to show each type of chromosome variants for comparison. In addition to show clear and representative images, these chromosomes used here were selected from other cells in considering the overall arrangements in a figure plate. For example, that M-type chromosome in 3a was not used as e1 chromosome because its coverage goes beyond the area limited in this plate, and chromosomes in the cell with e1 chromosome are widely spread, which cannot fit in the figure plate. That M-type chromosome in 3b is not used as e2 because overlapping with other chromosomes. That M-type chromosome in 3c is identical to e3 chromosome. The e4 chromosome with telomere signals indicates it as a complete chromosome after a segment deletion. 

Page 33, Fig. 3h-i: How do you know that these chromosomes are mini-chromosomes? They are not significantly different in size from other chromosomes.

Response: These chromosomes are meta centric type and shorter than other chromosomes, including M-, T-, A-type chromosomes, within the complement.

Page 34: Some of the pictures in Figures 5 and 6 are redundant, and the two pictures even including Figure 7 can be combined into a single Figure.

Response: We want to keep these three figures separated because the following reasons. 

Fig 5 shows the immuno-signals with different sizes and intensities at interphase nuclei; Fig 6 shows the distribution of centromeres at different mitotic stages detected by an anti-LaCENH3 antibody; Fig 7 shows the centromeres at individual chromosomes were specifically labeled by an anti-LaCENH3 antibody. All these images demonstrate the anti-LaCENH3 antibody is specific to centromere of M-,T-, A-type chromosomes and micronuclei of Lycoris.

Page 34, Fig. 6: 

Response: The typos have been corrected. (Line 723)

Page 35, Fig. 7: Please show complete cell image.

Response: The results of immunostaining experiments were consistently reproducible. However, we regret that we did not get a perfect image showing all chromosomes separated well from the results of more than ten times immunostaining experiments. In comparing with chromosome preparation for FISH analysis, it is hard in immunostaining experiments to observed complete cells with all of such huge chromosomes separated well within a field of view under high power object (63X) of the microscope. Instead, chromosomes in Fig 7 were collected from different slides to show specificity of anti-LaCENH3 antibody to centromeres at different type of chromosomes.

Page 35, Fig. 7: How do you determine if these signals are real or background signals?

Response: These are real signals because they are symmetrical signals at consistent position on these chromosomes. 

Page 35, Fig. 8: The typo “A” has been corrected to “An”. (Line 740)

Page 35, Fig. 8: Pay attention to grammar and Format.

Response: We have corrected this sentence (Lines 742-743)

Page 35, Fig. 8: (1) The same clone in different pictures had better display the same signal color. (2) Please arrange the images in the order of the numbers in the clone names (LaCenIP-1-LaCenIP-4). 

Response: The images in Fig 8 were rearranged based on reviewer's comments, as follows. In revised manuscript: 

1. Fig 8a and 8b are the same images as that in origin version. In both images, green signals indicate 5S rDNA and red signals indicate ChIP clones (Fig. 8, Lines 740-747).

2. Fig 8c-8f are grouped into Fig. 9 to show the similarity and different FISH signal patterns of four ChIP-clones. In these images, green signals indicate LaCenIP-1 and red signals indicate the other ChIP clone (Fig. 9, Lines 756-762). 

3. Fig. 9 and Fig. 10 in origin version are renumbered as Fig 10 and Fig 11, respectively.

4. Figure legends have been also revised. 

Page 35, Fig. 8: Are the same clones used in images d and f?

Response: Yes, both images (Fig. 9a and 9b in revised manuscript) were FISH results of same clones. The FISH signals in Fig 8f (Fig. 9b in revised manuscript) were over enhanced to show signals detected at mini-chromosome. 

A sentence, " The FISH signals were enhanced to present the signals at mini-chromosome", is inserted here to explain it. (Lines 759-761)

Page 35, Fig. 8f: micronucleus or mini-chromosomes?

Response: Because it is observed at metaphase, we change "micronucleus" to "mini-chromosome" (Fig. 9b, Line 760)

Page 36, Fig. 9: However, one chromosome indicated by arrowhead, on the right of the image Fig. 10 (a), shows clear red signals of LaCenIP-1R-E1 probe. 

Response: There are no arrowheads in Fig. 10a, did you mean the arrowhead in Fig 9(a)? The arrowheads in Fig 9a indicated T-type chromosomes. The centromere of T-type chromosome is known at the distal end of short arm, which has been visualized by immunostaining with anti-LaCENH3 antibody (as shown in Fig 7), but FISH signals of both LaCenIP-1R-E1 and LaCenIP-1 were not detected at the distal ends of each T-type chromosomes. 

Fig. 9 is renumbered to Fig. 10 in revised manuscript.

Page 36, Fig. 9: 

Response: The typo has been corrected. (Fig 10, Line 770)

Page 36, Fig. 10: Please show complete cell pictures of Fig. 10 (a) and (b). Figures 9 and 10 can be combined into a single Figure. 

Response: The images of the compete cells with FISH results of probes LaCenIP-1 and LaCenIP-1R-E1 are shown in Fig 10 (It was Fig 9 in origin version). Here we want to show the distribution patterns of FISH signals generated by both probes, which enable to identify individual chromosomes (Fig. 11 in revised manuscript). These chromosomes were collected from cells obtained from several FISH experiments (as shown in supporting information S2_Fig), which demonstrated these results are reliable and reproducible.

Responses to reviewer 2:

Page number, reviewer’s point marked on origin manuscript PDF: Our response (Line number in 'Revised Manuscript with Track Changes')

Page 9, Introduction (in origin manuscript, PDF): It was an editing error, “and” has been deleted (Line 62). 

Page 9, Introduction: It was a typo, “chromosomes” has been corrected to “chromosome” (Line 64).

Page 10, Introduction: “in situ” has been changed to “in situ”(Line 94).

---

## [Decision Letter · Decision Letter 1]

17 Sep 2021

Chromosomal variations of Lycoris species revealed by FISH with rDNAs and centromeric histone H3 variant associated DNAs

PONE-D-21-19380R1

Dear Dr. Chung,

We’re pleased to inform you that your manuscript has been judged scientifically suitable for publication and will be formally accepted for publication once it meets all outstanding technical requirements.

Kind regards,

Zhukuan Cheng

Academic Editor

PLOS ONE

Additional Editor Comments (optional):

Reviewers' comments:

Reviewer's Responses to Questions

**Comments to the Author**

1. If the authors have adequately addressed your comments raised in a previous round of review and you feel that this manuscript is now acceptable for publication, you may indicate that here to bypass the “Comments to the Author” section, enter your conflict of interest statement in the “Confidential to Editor” section, and submit your "Accept" recommendation.

Reviewer #1: All comments have been addressed

Reviewer #2: All comments have been addressed

2. Is the manuscript technically sound, and do the data support the conclusions?

Reviewer #1: Yes

Reviewer #2: Yes

3. Has the statistical analysis been performed appropriately and rigorously? 

Reviewer #1: N/A

Reviewer #2: Yes

4. Have the authors made all data underlying the findings in their manuscript fully available?

Reviewer #1: Yes

Reviewer #2: Yes

5. Is the manuscript presented in an intelligible fashion and written in standard English?

Reviewer #1: Yes

Reviewer #2: Yes

6. Review Comments to the Author

Reviewer #1: The authors have adequately addressed my comments raised in a previous round of review and I feel that this manuscript is now acceptable for publication. I have no additional comments to the author.

Reviewer #2: The authors have made essential revision according to my comments. The ms can be accepted for publication.

7. PLOS authors have the option to publish the peer review history of their article (what does this mean?). If published, this will include your full peer review and any attached files.

Reviewer #1: **Yes: **Yonghua Han

Reviewer #2: No

---

## [Editor Report · Acceptance letter]

23 Sep 2021

PONE-D-21-19380R1 

Chromosomal variations of *Lycoris* species revealed by FISH with rDNAs and centromeric histone H3 variant associated DNAs 

Dear Dr. Chung:

I'm pleased to inform you that your manuscript has been deemed suitable for publication in PLOS ONE. Congratulations! Your manuscript is now with our production department. 

Kind regards, 

on behalf of

Prof. Zhukuan Cheng 

Academic Editor

PLOS ONE